# Effectiveness of COVID-19 Vaccine Booster Shot Compared with Non-Booster: A Meta-Analysis

**DOI:** 10.3390/vaccines10091396

**Published:** 2022-08-25

**Authors:** Yajuan Zhu, Shuang Liu, Dingmei Zhang

**Affiliations:** 1Department of Epidemiology, School of Public Health, Sun Yat-sen University, Guangzhou 510080, China; 2NMPA Key Laboratory for Quality Monitoring and Evaluation of Vaccines and Biological Products, Guangzhou 510080, China

**Keywords:** COVID-19, vaccination, booster shot, Delta, Omicron

## Abstract

The analysis of the effectiveness of booster shots compared with primary vaccination is extremely vital. This paper aimed to summarize the results of all available evidence studies on the effectiveness of booster vaccination against severe acute respiratory syndrome coronavirus 2 (SARS-CoV-2). Articles published up to 21 June 2022 were systematically searched through PubMed and EMBASE databases. The searched studies were independently assessed for quality using the Newcastle–Ottawa Scale. Results: Seven studies (nine datasets) met the criteria and were included in this study. The pooled results demonstrated a 71% (OR = 0.29, 95% CI = 0.17–0.48) reduction in SARS-CoV-2 infection rates among subjects who received a booster shot compared with those who did not receive a booster shot of coronavirus disease (COVID-19) vaccine. In addition, this analysis emphasized that during the period when the Delta variant was predominant, subjects who received the booster shot showed an 82% (OR = 0.18, 95% CI = 0.13–0.25) reduction in infection rates. Moreover, during the period of dominance of the Omicron variant, subjects who received the booster vaccination displayed a 47% (OR = 0.53, 95% CI = 0.35–0.81) reduction in infection rates. This finding confirmed that booster vaccination against the Omicron variant is significantly less effective than that against the Delta variant. In pandemic periods, correlations between the dominant variant and the efficacy of the COVID-19 vaccine booster should be considered when making vaccine booster plans.

## 1. Introduction

Coronavirus disease (COVID-19), which is caused by severe acute respiratory syndrome coronavirus 2 (SARS-CoV-2), was declared a pandemic by the World Health Organization (WHO) on 11 March 2020 [1]. The pandemic has been persisting for more than two years since it started, seriously affecting the health of populations worldwide. However, no effective treatment for COVID-19 is available. Vaccination is considered an effective measure for the prevention and control of COVID-19 and plays a decisive role in the control of the global epidemic.

Although COVID-19 vaccination is fully implemented worldwide, SARS-CoV-2 remains rampant. As with other vaccines, the protective efficacy of the COVID-19 vaccine weakens over time after the completion of vaccination. A meta-analysis [2] showed that the vaccine effectiveness against SARS-CoV-2 infection declined by 21% from one month to six months after primary vaccination. Following booster vaccination, however, a higher level of antibodies will be produced in the recipient’s body, thus providing protection against the virus [3,4]. Therefore, booster doses are currently being administered in various countries to improve the overall immunization level of populations. 

The most notable feature of the epidemic is the constant and rapid mutation of SARS-CoV-2. As of 13 August 2022, WHO has identified five variants of concern (VOCs), namely Alpha, identified on 18 December 2020; Beta, identified on 18 December 2020; Gamma, identified on 11 January 2021; Delta, identified on 11 May 2021; and Omicron, identified on 26 November 2021 [5]. Currently, the dominant strain shaping the global outbreak has changed from the Delta variant to Omicron. Compared with the previously prevalent Delta variant, Omicron has more key mutations, including up to 32 mutations in the spike protein, several of which may be associated with immune escape and higher infectiousness. Nevertheless, on the whole, the Omicron variant causes fewer symptoms and significantly fewer cases of severe hospitalization or death than the previous variants. The booster shot has been shown to be effective in current studies. According to the results of domestic and international studies [6,7,8], COVID-19 vaccination significantly reduces the rate of infection, hospitalization, and population mortality caused by SARS-CoV-2 variants and contributes significantly to the prevention and control of outbreaks. Vaccine effectiveness against symptomatic infections caused by omicron was approximately 50% in the first three months after the second dose of vaccine, but vaccine effectiveness against hospitalization and death due to Omicron infection remained high at over 70% after the second dose and above 90% after the booster dose. Overall, the current vaccination still provides protection against the variants.

The effectiveness of a third dose (booster) compared with the two doses in a real-world setting is unknown, and a meta-analysis of the effectiveness of real-world-based booster shots compared with primary vaccination is extremely important. Concerns regarding the possible lower vaccine effectiveness against the Omicron variant compared with that against the Delta variant have emerged. Whether a difference in vaccine effectiveness between the booster doses during the Omicron- and Delta-dominant period should be determined to address both questions. Therefore, this paper aimed to conduct a meta-analysis to summarize the results of all available evidence and studies on the effectiveness of booster vaccination compared with primary vaccination against SARS-CoV-2.

## 2. Materials and Methods

### 2.1. Search Strategy

This meta-analysis was conducted in accordance with the Preferred Reporting Items for Systematic Reviews and Meta-Analyses guidelines. This study was prospectively registered on PROSPERO (n.CRD42022337356). Articles published up to 21 June 2022 were systematically searched through PubMed and EMBASE databases. The search keywords included (“SARS-CoV-2” or “COVID-19” or “2019-nCov”) AND (“Vaccine” or “Vaccination”) AND (“booster” or “booster shot” or “third dose” or “additional dose”) AND (“Efficacy” or “Effectiveness”).

### 2.2. Inclusion and Exclusion Criteria

The inclusion criteria included (1) studies with outcomes of COVID-19 infection; (2) cases defined as individuals who tested positive for SARS-CoV-2 as confirmed by polymerase chain reaction, (3) observational studies, (4) studies published in English; (5) studies that excluded patients who had been infected with COVID-19 prior to their first vaccination.

The exclusion criteria included (1) articles for which the full text could not be found; (2) systematic reviews, commentaries, case reports, letters, and guidelines; (3) studies of non-human subjects, (4) articles for which data could not be extracted; (5) studies considering only hospitalization, serious illness, death and other serious outcomes without considering SARS-CoV-2 infection.

### 2.3. Data Extraction and Study Quality Assessment

Data extraction and quality assessment were carried out independently by two authors, with disagreements identified by a third author. The following data were extracted from the included articles: name of first author, country, study design, vaccine, dominant variant, age range, and the number of booster and non-booster vaccinations in cases and controls. Table 1 summarizes the extracted data.

The included studies were independently assessed for quality using the Newcastle–Ottawa Scale (NOS), which was designed for observational and non-randomized studies. Scores of 0–3, 4–6, and over 7 stars were considered of low, moderate, and high quality, respectively.

### 2.4. Data Synthesis and Analysis

The *I*^2^ statistic was applied to assess the heterogeneity between studies. If *I*^2^ > 50%, which indicates a high heterogeneity, the random-effects model was recommended. Meanwhile, if *I*^2^ < 50%, the fixed effects model was normally employed for the analysis. Pooled odds ratio (OR) and 95% confidence interval (95% CI) estimates were calculated to determine the association between COVID-19 vaccine booster shots and infection. Dominant variants were considered for subgroup analysis to further identify the sources of heterogeneity. A two-tailed *p* < 0.05 was deemed to be statistically significant. Statistical analyses were conducted by STATA/SE version 16.0 (StataCorp, College Station, TX, USA).

## 3. Results

### 3.1. Literature Retrieval and Literature Quality Evaluation

A total of 904 records were identified from the literature search. After removing duplicates, 578 articles were identified and subsequently screened by title, abstract, and full text. After excluding the ineligible studies from the title or abstract screening, 35 studies were reviewed in full text. As a result, seven studies (nine datasets) met the criteria and were included in this study. Studies reporting the efficacy of different dominant variants were treated as a separate dataset in the meta-analysis (Figure 1).

Table 1 shows details on the included studies (*n* = 7). Most of the included research (*n* = 4) was conducted in the United States. The meta-analysis included 5,510,606 subjects, of whom 867,361 were infected patients. Of the seven studies, one was a retrospective cohort research study, and the rest were test-negative design studies. All of these studies were published in 2022, and most were conducted on mRNA vaccines. In addition, the quality of all included studies was assessed with the NOS. Four and three studies showed moderate and high quality, respectively.

### 3.2. Effectiveness of COVID-19 Vaccine Booster versus Non-Booster Doses

Based on the degree of heterogeneity of the included studies (*I*^2^ > 50%), the correlation between COVID-19 vaccine booster shots and infections was analyzed using a random-effects model. The pooled results revealed that the COVID-19 vaccine booster was a protective factor against infection relative to the non-booster doses (OR = 0.29, 95% CI = 0.17–0.48). A subgroup analysis was subsequently conducted based on the dominant variant strains: the Delta- and Omicron-dominant phase groups (Figure 2). This analysis emphasized that booster vaccination during the Delta variant-dominant period was more effective in preventing COVID-19 disease compared with the Omicron variant-dominant period (*p* < 0.001). During the dominance period of the Delta variant, the booster-vaccinated subjects demonstrated a significant reduction in infection rates compared with non-booster-vaccinated subjects (OR = 0.18, 95% CI = 0.13–0.25). During the period of dominance of the Omicron variant, booster-vaccinated subjects also displayed a reduction in infection rates compared with those who did not receive the booster vaccine (OR = 0.53, 95% CI 0.35–0.81). This finding supported the effect of different variant strains on the effectiveness of booster vaccination.

### 3.3. Publication Bias and Sensitivity Analysis

The use of funnel plots and Egger’s test for publication bias assessment was not feasible on account of the small number of studies included in the pooled analysis (*n* < 10). Sensitivity analysis by excluding one study at a time had no significant effect on the results (Figure 3, Figure 4 and Figure 5).

## 4. Discussion

The COVID-19 vaccine booster provides a further enhancement or restores protection that may have waned over time after the initial series of vaccinations. Furthermore, the booster is one of the most essential means of defending individuals from serious illness or death attributed to COVID-19 [16,17]. In addition, the booster dose is recommended for anyone eligible for vaccination given the presence of the more infectious variants of Delta and Omicron. A systematic review [18] summarized studies associated with the efficacy of the booster dose against the Omicron variant, and all the results reviewed supported the evidence for the efficacy of the booster dose vaccine against SARS-CoV-2 variants, including Omicron.

Vaccine effectiveness is an indicator that measures the effectiveness of immunization in protecting people from consequences, such as infections, symptomatic illnesses, hospitalization, and death. This meta-analysis focused on examining the efficacy and effectiveness of the COVID-19 vaccine booster compared with non-booster doses in reducing infection rates to provide strong evidence for health policy makers in response to ongoing pandemics. Despite the considerable heterogeneity between studies, our findings provide evidence that the booster was less effective against the Omicron variant strain than the Delta variant. Overall, booster vaccination against COVID-19 was effective in reducing the number of COVID-19 cases.

This meta-analysis is the first to report the pooled data on the effectiveness of the COVID-19 vaccine booster shot in comparison with the non-booster shot. Of the included studies, apart from one research study that did not specify an age range, all covered young people and older people aged 18 years and over. Furthermore, the sample sizes of the individual studies ranged from several thousands to several millions. One study was retrospective cohort research, and the other six applied a test-negative design, which is commonly used in vaccine efficacy studies. The results of the studies included in the meta-analysis were consistent. The pooled results demonstrated a 71% (OR = 0.29, 95% CI = 0.17–0.48) reduction in SARS-CoV-2 infection rates among subjects who received a booster shot compared with those who did not receive a booster shot of COVID-19 vaccine. Moreover, this analysis emphasized that during the period when the Delta variant was predominant, subjects who received the booster shot showed an 82% (OR = 0.18, 95% CI = 0.13–0.25) reduction in infection rates. In addition, during the period of dominance of the Omicron variant, subjects who received the booster vaccination displayed a 47% (OR = 0.53, 95% CI= 0.35–0.81) reduction in infection rates. This finding supported the effect of the different variants on the effectiveness of booster vaccination. Booster vaccination during the Delta variant-dominant period was more effective in preventing COVID-19 disease than the booster vaccination during the dominant period of the Omicron variant (*I*^2^ = 100.0%, *p* < 0.001). This result confirmed that booster vaccination against the Omicron variant was significantly less effective than that against the Delta variant. Compared with the Delta variant, the Omicron variant has a larger area of variation, and its “face” is largely different and more infectious than the original strain [19,20,21]. To better control the pandemic, experts must fundamentally redesign the booster vaccine to specifically target the Omicron variant in the new context. At present, a number of companies have developed new generations of COVID-19 vaccines against the Omicron antigen and are carrying out clinical trials.

The present meta-analysis had several limitations. In particular, the analysis of between-study heterogeneity and subgroup analyses failed to reduce the overall heterogeneity. The use of funnel plots and Egger’s test for publication bias assessment was not feasible on account of the small number of studies included in the pooled analysis (*n* < 10). In addition, given the sparse literature, no distinction was made between different age groups, and data on the effectiveness in preventing severe infections were not available. The potential differences in effectiveness of preventing infection or severe infections between populations could not be compared. Focus on the discrepancies between homologous and heterologous reinforcements was also lacking. Also, in the absence of data, whether there were potential differences associated with the type of vaccine used for the first two doses and for the booster dose was not analyzed. Furthermore, the level of epidemiological dynamics and vaccine policies in each country could also affect the assessment of the effectiveness of booster doses. The major weakness was that the booster shot was administered later compared with the non-booster shot, and the effect of time on the effectiveness of the booster shot was not ruled out. Similarly, in comparison with the Delta strain, the Omicron strain emerged later with an accompanying tendency for the antibodies to fade away. Duration is the most critical confounding factor. Although the time confounder cannot be ruled out, public vaccination did not take place at the same time, and some people were vaccinated after the Omicron epidemic. Therefore, the time of infection with the Delta strain after vaccination and the time of infection with Omicron after vaccination were not totally incomparable. It is merely that information on time was not available due to data limitations. In the same studies included in the meta-analysis, the time of cases and controls were comparable, and hence they were somewhat comparable in the time of cases and controls across the meta. Therefore, despite the fact that time was the most critical confounding factor, the conclusion of this article could provide some suggestive points. More detailed information on the COVID-19 vaccine booster and all the potential factors that can affect the outcome are worthy of attention.

In summary, preliminary data on the COVID-19 vaccine booster suggested that potential dominant strain differences occur in terms of the efficacy. In pandemic periods, correlations between the dominant variant and the efficacy of the COVID-19 vaccine booster should be considered when making vaccine booster plans.

## 5. Conclusions

This meta-analysis summarized the efficacy of the COVID-19 vaccine booster versus non-booster in reducing infection rates, and the findings supported the comparable effectiveness of the booster. In addition, the booster was less effective against the Omicron variant than the Delta variant. Further studies on the effectiveness of real-world vaccine booster shots are encouraged to explore other sources of heterogeneity that may influence the efficacy of meta-analysis results.

## Figures and Tables

**Figure 1 vaccines-10-01396-f001:**
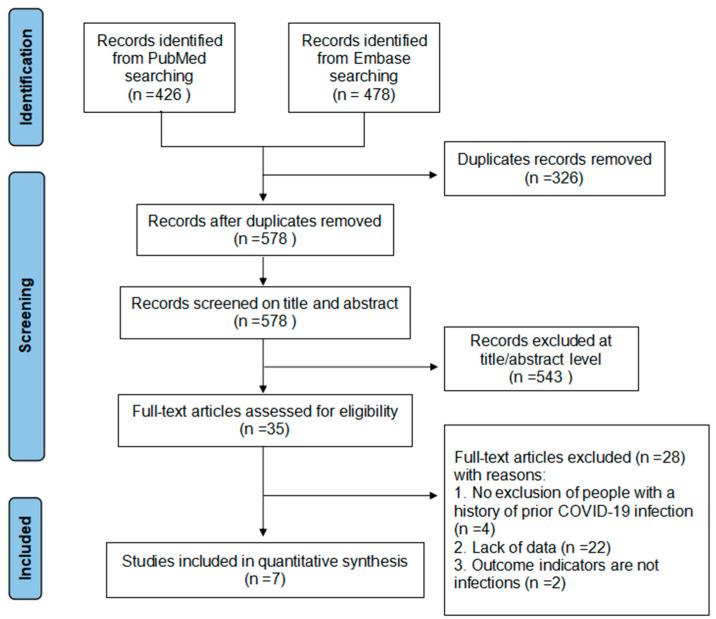
Flow chart of literature search and screening.

**Figure 2 vaccines-10-01396-f002:**
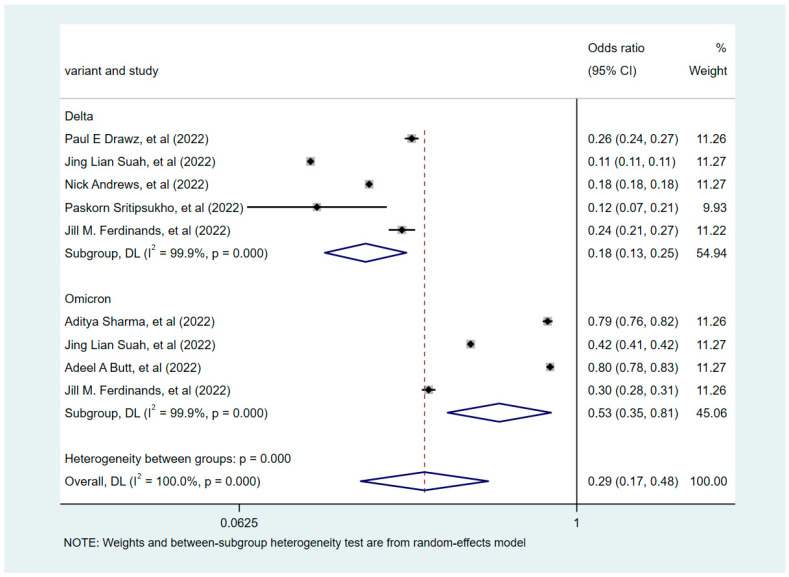
Forest plots showing pooled risk ratio of COVID-19 vaccine booster shots associated with infections.

**Figure 3 vaccines-10-01396-f003:**
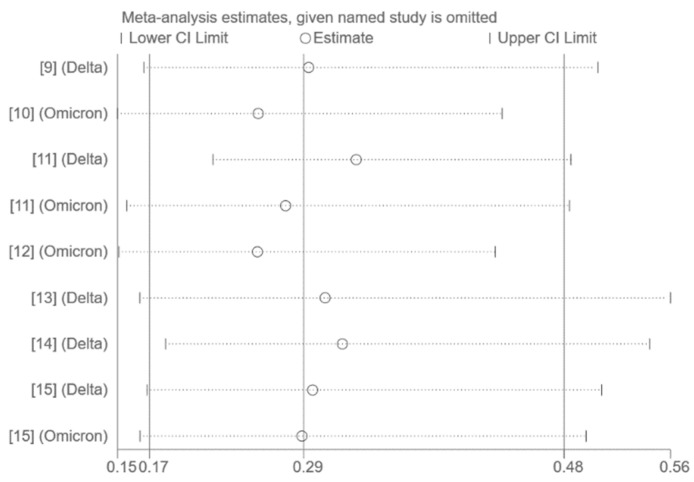
Sensitivity analysis (all publications) [9,10,11,12,13,14,15].

**Figure 4 vaccines-10-01396-f004:**
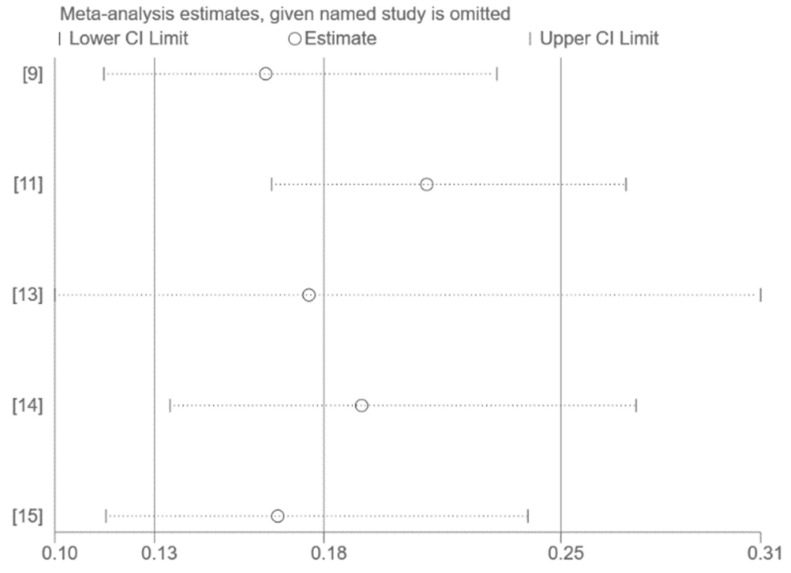
Sensitivity analysis (only Delta) [9,11,13,14,15].

**Figure 5 vaccines-10-01396-f005:**
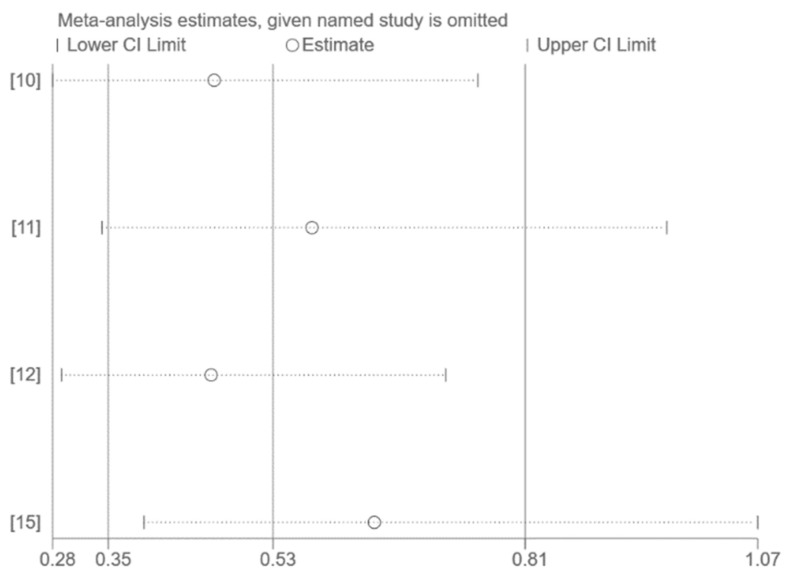
Sensitivity analysis (only Omicron) [10,11,12,15].

**Table 1 vaccines-10-01396-t001:** Characteristics of the included studies.

Study Author, Year	Country	Study Type	Vaccine	Dominant Variant	Study Periods	Age Range	Booster		No booster		Outcome	NOS
							Positive	Negative	Positive	Negative		
Paul E Drawz, et al. (2022) [9]	United States	Test-negative case–control design	BNT162b2 or mRNA-1273	Delta	From 29 August 2021 to 27 November 2021	≥19	1347	46,526	15,702	139,718	Infection	6
Aditya Sharma, et al. (2022) [10]	United States	Test-negative case–control design	BNT162b2 or mRNA-1273	Omicron	From 1 December 2021 to 12 March 2022	NA	4226	404,548	5356	403,418	Infection	8
Jing Lian Suah, et al. (2022) [11]	Malaysia	Test-negative case–control design	BNT162b2 or CoronaVac or AZD1222	Delta	From 27 October 2021 to 4 February 2022	≥18	38567	882,109	280,560	720,077	Infection	8
Jing Lian Suah, et al. (2022) [11]	Malaysia	Test-negative case–control design	BNT162b2 or CoronaVac or AZD1222	Omicron	From 5 February 2022 to 22 February 2022	≥18	135,425	424,968	171,058	224,378	Infection	8
Adeel A Butt, et al. (2022) [12]	United States	Retrospective cohort study	BNT162b2 or mRNA-1273	Omicron	From 1 January 2022 to 20 February 2022	≥21	8444	454,506	10,462	452,488	Infection	5
Nick Andrews, et al. (2022) [13]	England	Test-negative case–control design	ChAdOx1-S or BNT162b2	Delta	From 13 September 2021 to 5 November 2021	≥18	17,655	143,001	159,593	234,684	Symptom	8
Paskorn Sritipsukho, et al. (2022) [14]	Thailand	Test-negative case–control design	CoronaVac or BNT162b2 or ChAdOx1-S	Delta	From 25 July 2021 to 23 October 2021	≥18	13	478	181	787	Infection	5
Jill M. Ferdinands, et al. (2022) [15]	United States	Test-negative case–control design	mRNA vaccine	Delta	From 26 August 2021 to 22 January 2022 *	≥18	347	13,860	8136	77,235	Infection	6
Jill M. Ferdinands, et al. (2022) [15]	United States	Test-negative case–control design	mRNA vaccine	Omicron	From 26 August 2021 to 22 January 2022 *	≥18	1938	8993	8351	11,471	Infection	6

Note: * In the original tables of the included study, the authors only distinguished between delta-dominated periods and omicron-dominated periods, without specifying the time points of the two periods.

## Data Availability

Data supporting the reported results are available on request to the Authors.

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
