# Peer review of "Effectiveness of COVID-19 Vaccine Booster Shot Compared with Non-Booster: A Meta-Analysis"

_vaccines, 2022, doi:10.3390/vaccines10091396_

Round 1

Reviewer 1 Report

Really interesting article providing relevant information which deserves to be published

Only a few comments/suggestions below:

Abstract =

·       Should be slightly clarified, in particular regarding the outcome which is called “increase in the protection against SARS-CoV-2 infection” (line 15, likely the worse one), “reduced infection” (line 19), “lower infection rate” (line 21). This should be homogeneized. The term “reduction in infections rates” often used in the text could also be used.

Introduction,

·       page 2, lines 54-55: the following sequence is unclear or inexact “A meta-analysis of the effectiveness of real-world based booster shots compared with primary vaccination is extremely important. The effectiveness of a third dose (booster) compared with the two doses is unknown.” Should the author refer only to meta-analyses in a real word setting? Please clarify

Results and discussion :

·       Main point: no data on the effectiveness in preventing severe infections are provided. If it is not possible, this should be mentioned and in all cases, the potential difference (or not) of effectiveness in preventing infection on one hand and severe infections on the other hand should be discussed (in particular in populations at high risk)

·       whether there could be potential differences linked to the type of vaccine used for the first two doses and for the booster dose is not analyzed or at least discussed

·       no information is provided on the dynamics levels of the epidemics and on vaccine policy in the countries (in particular in the USA 5/8 studies) at the time the studies were conducted, and how it could affect the evaluation of the effectiveness of the booster dose (apart from “the effect of time on the effectiveness of the booster shot was not ruled out”, line 197)

Author Response

Please see the attachment, thank you very much.

Reviewer 2 Report

In this review author analyze the effectiveness of a vaccine shot compared with non-booster patients making a difference regarding type of SARS-CoV-2  variant (Delta and Omicron)

Major points:

Comment 1: Authors infer from the analysis that the vaccine effectiveness (VE) will depend different variant strains (82% [95% C.I. 75- 87] for delta variant and 47% (95% C.I.  19-65) for Omicron variant) and one of the conclusions is “In addition, the booster was less effective against the Omicron variant than the Delta variant”. However this statement is probably biased because Delta and Omicron variants barely coexist in time: OMS assign the category of Variant Of Concern (VOC) to Delta variant at 11-05-2021, meanwhile this status was assigned to Omicron variant at 26-11-2021 differing more than half a year (https://www.who.int/activities/tracking-SARS-CoV-2-variants). In Discussion section author mentioned “Similarly, in comparison with the Delta strain, the Omicron strain emerged later with an accompanying tendency for the antibodies to fade away. Duration is the most critical confounding factor.”. The timing since vaccine booster is essential to compare VE of different analyzed strains, otherwise the difference could be associated to a decreasing effect though time. Authors mentioned that the lost protection  with time from initial vaccination (L147-L148), and this phenomenon probably happens  also with booster dose. This would imply that the main conclusion of this review would be  completely biased.

Comment 2: SARS-CoV-2 infection is a key outcome related to COVID-19; however,  I miss analysis regarding to hospitalization, ICU admission or deaths. The analysis of vaccine effectiveness  cannot be understood without these three outcomes. Vaccination effectiveness related to infection seems to be higher  for Delta than for Omicron strain (according authors’ conclusions); however, it is possible that final evolution of COVID-19 would be more severe for Delta than for Omicron strain. In summary, although the protection against Omicron infection may be lower, the course of the disease may be milder, so the general computation could be that the booster dose works better against the Omicron variant. I suggest to remove the exclusion criteria “studies with  outcomes of hospitalization, serious illness, death, and other serious outcomes”, and to performed the analogous analysis.

Minor comments

Comment 3: The screening by title and/or abstract could be too strict. For instance, the study Barda et al (Lancet, 2021), cited in the present manuscript, should be included in this meta-analysis, because although,  it is not mentioned in the abstract, they consider SARS-CoV-2 infection as secondary outcome.

Comment 4: Figure 3 need to be improved, there are number and labels overlapped. Moreover, authors obtained three estimations (all publications, only Delta, and only Omicron), but only shows the sensitivity of all publications. 

Comment 5: Authors need to extend the information displayed about VOCs.

Comment 6: In the line of Comment 1 authors have to add study periods to Table 1.

Comment 7: In methods section authors mentioned “Potential publication bias was assessed using funnel plots and Egger’s test.”; however, in results section author also said “The use of funnel plots and Egger’s test for publication bias assessment was not feasible on account of the small number of studies included in the pooled analysis (n < 10).”. The lack of publication analysis bias should be included as a limitation, and the mention in methods section should be removed.

Comment 7: Although I suggest to remove the exclusion criteria “studies with  outcomes of hospitalization, serious illness, death, and other serious outcomes” (previous comment), I imagine that this condition refers to remove studies that only consider these outcomes without considering SARS-CoV-2 infection. For instance, Butt et al. (2022) makes the next statement “We assessed 3 outcomes: confirmed infection (defined as a PCR-positive swab specimen), hospitalization with coronavirus disease 2019 (COVID-19) (within 14 days after a positive test for SARS-COV-2), and intensive care unit (ICU) admission or fatal COVID-19”, and how it is expressed in text, this study should be removed. This should clarified. 

Author Response

(The authors gave the same response as above.)

Round 2

Reviewer 2 Report

I am glad that the author has extended the limitations section according to my comments. 

· To fully understand the results displayed in the analysis, I have some concerns about the displayed results. It is not clear which results were used from each study. For instance, the paper Shua et al. displayed 8 booster vaccine effectiveness estimations, depending on the type of vaccine combination. I suppose that results refers to adjusted  vaccine effectiveness for PPP results (3×BNT162b2) in comparison with PP (2×BNT162b2); however the sample sized displayed correspond  to the sum of all combinations. This can be quite confusing, because the displayed OR were not obtained with the mentioned data. Furthermore, the Odds Ratio associated with Booster dose for adjusted model and Omicron variant does not match with the results displayed. The Odds Ratio related to Sharma et al. study is 0.79 (0.76-0.82) (Figure 2) which corresponds with a vaccine effectiveness of 21% (18-24); however  the text mention (Table 2)  that vaccine effectiveness  for third dose in comparison with primary  series is 30.1 (26.2–33.7) for BNT162b2 vaccine dose and 37.1 (32.2–41.7) for mRNA-1273 booster dose. There are similar concerns related to  (Andrew et al.  Drawz et al., and Ferdinands et al., etc). Authors should specify which estimations were considered  to perform this meta-analysis.

·  Please change “As of August 13, 2022, WHO has identified five variants of concern (VOCs), namely Alpha, identified on 18 December 2020 , Beta, identified on 18 December 2020 , Gamma, identified on 11 January 2021 , Delta, identified on 11 May 2021 and Omicron, identified on 26 November 2021” by “As of August 13, 2022, WHO has identified five variants of concern (VOCs), namely Alpha, identified on 18 December 2020; Beta, identified on 18 December 2020; Gamma, identified on 11 January 2021; Delta, identified on 11 May 2021; and Omicron, identified on 26 November 2021”

·  L200: What was the statistic employed to analyze the statistical differences between booster vaccine effectiveness for Delta and Omicron variants.

·  The Odds Ratio related to Butt et al. is 0.81 (from Table 2, RVE against infection), not 0.80.

·  Figures 3 to 5: authors should increase the x-axis plot-limits, because there are boundaries which do not  fall into the margin plot. 

·  In Figure 3 there is two x-labels overlapped.

·  In Figures 4 and 5, the strain variants next to the publication  label are not necessary.

·  L226: Please change “omicron” by “Omicron”

·  L228: Add “ ” after the dot.
